# Free Amino Acid Content in Human Milk Is Associated with Infant Gender and Weight Gain during the First Four Months of Lactation

**DOI:** 10.3390/nu11092239

**Published:** 2019-09-17

**Authors:** Manuel E. Baldeón, Federico Zertuche, Nancy Flores, Marco Fornasini

**Affiliations:** Centro de Investigación Biomédica, Facultad de Ciencias de la Salud Eugenio Espejo, Universidad UTE, Quito 170147, Ecuador; federico.zertuche@ute.edu.ec (F.Z.); nancilin@hotmail.com (N.F.); marco.fornasini@ute.edu.ec (M.F.)

**Keywords:** human milk, breastfeeding, gender, growth, free amino acids, branched chain amino acids, glutamate, glutamine

## Abstract

Background: There is a growing interest regarding the physiological role of free amino acids (FAA) present in human milk (HM). Recent studies show FAA in HM could be influenced by infants’ gender and could have an important role in their growth and development. We studied the concentrations of FAA in HM and potential associations with infants’ gender and their patterns of growth in a cohort of Ecuadorian women. Methods: Human milk samples were collected after approximately eight hours of overnight fast within one week (colostrum), 2 weeks (transition milk), and 2 and/or 4 months (mature milk) after parturition. Free AA were determined by cation-exchange chromatography separation. Results: We observed significantly higher concentrations of Glu 14.40 (1.35, 27.44), Gly 1.82 (0.24, 3.4), Cys 0.36 (0.03, 0.68), and Tyr 0.24 (0.02, 0.46) in HM intended for boys. Free Glu, Gly, Cys, and Tyr concentrations increased with time of lactation. In addition, there were higher concentrations of Glu 28.62 (1.78, 55.46) and Ala 7.16 (1.26, 13.06) in HM for children that presented faster weight gain than for those with slower gain. Conclusions: The present results showed that there are differences in FAA levels in HM intended for male and fast-growing children.

## 1. Introduction

Breast milk is a complex matrix of macro- and micronutrients, hormones, immune components, and other metabolites including free amino acids (FAA) that support the growth and development of children [1]. Reports from diverse ethnic groups around the world indicate that FAA content in human milk (HM) is similar and perhaps reflects an evolutionarily conserved trait [2]. Due to its unique composition and physiological properties, current recommendations for optimal child growth and development is exclusive breastfeeding during the first six months of life [1]. Children that are not breastfed or have been breastfed for shorter periods of time present greater risk of obesity during childhood [3]; obesity is associated with chronic metabolic, immune, and behavioral diseases [4,5,6,7]. Studies on individual breast milk components throughout lactation have contributed to our understanding of the mechanisms that relate breast milk with infant growth, development, and chronic diseases.

Current evidence indicates that FAA in HM play an important role in infant growth and development [8,9]. Thus, branched-chain and insulino-trophic amino acids (BCAA and ITAA) are more abundant in breast-milk associated with faster infant growth [9]; also, free glutamate, glutamine, cysteine, and glycine favor gastrointestinal [10,11] and immune system development and maturation [12,13].

Previous work shows that FAA content in HM could be related to the gender of the infant [14]. In that study, measurements of FAA in mature milk collected monthly for 3 months after delivery show higher concentration of free glutamine associated with male infant gender [14]. In addition, metabolomic studies on human milk content show that there is a positive association between BCAA (leucine, isoleucine, and valine) and ITAA (leucine, isoleucine, valine, threonine and arginine) concentrations and infant growth rate [9]. We previously reported the patterns of FAA concentrations in colostrum, transition, and mature milk of 65 primiparous Ecuadorian women aged 14 to 27 years [15]. In that study, we observed a steady increase with time of taurine, glutamic acid, glutamine, and alanine throughout the first four months of lactation [15]. Here we complement our previous work and report the relationships between FAA content in human milk of Ecuadorian women (colostrum, sample 1-S1; transition S2, and mature breast milk S3) and their children’s gender and pattern of growth.

## 2. Materials and Methods

### 2.1. Subjects, Material and Methods

This prospective cohort study included 65 lactating mothers between 14 and 27 years of age that provided colostrum, transition, and mature milk during the first 4-months of lactation to determine FAA content; mothers and their children were recruited between 2009 and 2011.

Participating mothers were recruited after their child’s delivery at Hospital Gíneco-Obstétrico Isidro Ayora in Quito, which serves low-income families. We invited to participate 65 healthy consecutive primiparous women from Quito who gave birth to a single, at-term healthy infant without obstetric complications, and who indicated their willingness to feed their child exclusively with their breast-milk for at least 4 months. Women with a history of abortions or infections during pregnancy were not included in the study [15].

### 2.2. Breast Milk Collection and Amino Acids Measurements

Breast milk was collected by participating women using a manual breast pump (Camera M-11133, Quito, Ecuador) that dispensed milk into a 50 mL aseptic tube (Falcon, Quito, Ecuador); women were asked to wipe the breast with a clean warm cloth and sample 10 mL of milk from each breast. Samples were transported to the laboratory at 4 °C within 2-h of collection. Upon arrival, samples were deproteinated with 6% 5-sulfosalicylic acid dehydrated, and centrifuged at 3000 rpm for 15 min; serum supernatants were filtered with a 0.45-μm filter (Whatman 25mm GD/X) and stored in 5 mL cryogenic tubes in liquid nitrogen until amino acid determination at Ajinomoto Co. Inc., Institute of Life Science, Japan [15]. Breast milk samples were collected between 06:00–08:00 am, after approximately 10 h of fasting; colostrum—S1 was collected within the first week postpartum; transition milk—S2 between 13 to 17 days postpartum. In addition, two-mature milk samples were collected at 2 —S3 and/or 4 months postpartum—S4. Before AA measurements, samples were cleared with an Amicon Ultra centrifugal filter (Millipore, Tokyo, Japan) to remove molecules >10 kDa. Amino acid determination was performed as previously reported [15,16]. Free amino acid concentrations were measured in μmol/dL.

### 2.3. Anthropometric Measurements

Weight and height were taken with mothers wearing light clothing and no shoes by trained personnel. A standardized clinic weight-height scale (Seca clara 803, Hamburg, Germany) was used to determine weight and height of each mother at the time of recruitment and then weight at each subsequent visit. To record the children’s weight, first the mother was weighed alone, the weight was recorded and subsequently the mother was weighed again holding her child wearing no clothes; the difference in weight between both measurements was registered as the weight of the child. Head circumference was measured with a non-distensible tape that was placed over the most prominent part of the occiput and around the forehead. Head circumference was recorded to the nearest 0.1 cm at baseline, and then at each subsequent visit. At the time of recruitment, all but 10 of the children and all but three of the women were measured.

### 2.4. Ranking of Infants according to Growth Rate

Infant population was divided in fast, intermediate, and slow growers using weight and head circumference at 1, 2, 8, and 16 weeks to estimate a slope and an intercept for each of the children. Then, the slopes and the intercepts were divided in tertiles. Children with a slope in the highest tertile were considered fast growers. This approach allowed assessment of the individual concentrations of FAA and the speed of child growth.

At the same time that the anthropometric measurements were taken, human milk samples were also collected, S1 to S4. To have a precise estimate of the speed of growth, we only used data from subjects whose FAA concentrations were measured at the four time points of the study.

### 2.5. Statistical Analyses

Descriptive statistics such as frequencies, percentages, means, or medians with their respective measures of dispersion were used to describe the data. To assess differences between groups in continuous variables we used ANOVA or Kruskal Wallis. A *p* value < 0.05 was considered statistically significant.

#### Regression Models

We used simple linear models to summarize differences in the concentrations of FAA over time between HM for boys and girls to summarize that the concentration pattern for one gender is above or below the other.

For the most basic model we used:
*FAA* = α0 + α1*week*+ α2 *sex* + αid
where FAA: is the free amino acid concentration; week: is the week at which samples were taken, sex: is the baby’s gender, and the last term represents individual effects.

The sex coefficient of this simple regression model accounts for the difference between the concentrations for boys and girls after controlling for time and individual effects.

We used a multilevel regression model because these multilevel models give conservative estimations of the differences in the concentration of the FAA by sex since these models take into account possible unobserved individual effects.

The multi-level regressions and plots were produced in R V3.6.0 using the lem4 and tidyverse packages [17]. All the code and data for the regression analysis are available at https://github.com/notblank/AA-leche-humana.

Analysis was performed with SPSS V25 (IBM Corp. Released 2018. IBM SPSS Statistics for Windows, Version 25.0. Armonk, NY, USA: IBM Corp.) and r software [18].

### 2.6. Ethics

The study was approved by the Human Subjects Protection Committee at Universidad San Francisco de Quito, and informed consent was obtained from each participating mother; Ethical Approval Number: 2009-2.

## 3. Results

### Characteristics of Study Population

During the study some patients missed collection dates (colostrum, sample 1-S1; transition S2, and mature breast milk S3 and S4). For those patients and time points of collection, data are missing. There were 50 mother infant pairs with complete anthropometric measurements at the beginning of the study. There were 15 mother infant pairs with missing data because anthropometry measurements were not performed or the sex of the infant was not recorded at base line.

In relation to milk samples, Table 1 summarizes the number of milk samples analyzed for each time point of collection. There were at least 37 observations used to fit the regression models (see below). To address the missing data, we compared basal anthropometric measurements of mothers and children with complete and incomplete data and there were not statistically differences between the groups. In addition, since we were interested in the difference in the concentration of FAA by sex, we analyzed the sex distribution for patients with missing and with complete data for each time point. After performing a Fisher exact test and a Score type-test we found that there were no differences in the analyzed groups with or without missing data. Together this analysis demonstrated that the two groups are comparable and no bias was introduced considering data missing at random. The details of the computations and the corresponding data and code can be found in the following link: https://github.com/notblank/AA-leche-humana/blob/master/Missing%20at%20random.ipynb”.

Table 2 shows the demographic and anthropometric characteristics of participating populations. Adolescent and adult mothers participated in this trial; none of them presented malnutrition at the time of recruitment. According to World Health Organization standards, on the first week after birth, 2% of children had abnormal low head circumference values; and 8% of children had low weight. While at week 16, 13.8% of children had abnormal low head circumference; and all infants had normal weights. As expected, male infants gained more weight and head circumference than female infants, Table 2. There were no statistically significant differences for weight and head circumferences between male and female infants at one week after birth (weight, *p* = 0.096; head circumference *p* = 0.349); at 16 weeks of age there were no differences in children’s weight (weight, *p* = 0.433); however, there were statistically significant differences in head circumference (*p* = 0.011), Table 2.

#### Differences in Free Amino Acid Concentrations in Human Milk for Female and Male Infants

We compared the concentration of FAA in HM for male and female infants in our study group; we observed statistically significant higher concentrations of Glu, Gly, Cys, and Tyr and a trend for higher levels for free Gln and Ala in HM intended for boys, Figure 1A,B. Figure 2 shows the pattern of Glu, Gly, Cys, and Tyr during the first- 4 months of lactation (colostrum, transition, and mature milk) for both genders. Free Glu, Gly, and Cys increased with the time of lactation and the concentrations were significantly higher in HM for males than for females. A similar pattern was observed for Tyr levels; however, the pattern of Tyr levels for girls was more irregular, Figure 2. A further analysis using the same multilevel model as before but with the added interaction between sex and time of lactation indicated that although the concentrations of most FAA for boys were greater overall, the speed at which the concentrations increased over time was similar for both genders. Figure 2 shows that the concentrations of FAA over time were higher for boys than for girls. On the other hand, there were no important differences in essential FAA concentrations in HM for females or males, Figure 1B.

Previous works report that concentrations of BCAAs in HM were greater in fast growing children [9]; consequently, it was also important to compare the concentrations of FAA considering the pattern of growth in the present study. Our infant population was initially divided in fast, intermediate, and slow growers based on the tertiles of growth of weight and head circumference during the four months of lactation, Figure 3A,B. Children that were born with higher weight and head circumference grew faster than children born with lower anthropometric values, Figure 3. When we used the tertiles related to weight we observed a statistically significant difference in GLU and ALA concentrations in favor of fast-growing children, Figure 4A,B. Also, the differences in GLN concentrations were greater in fast growing children than in slow growers, although the differences were not statistically significant. When we analyzed head circumference growth, the differences in the pattern of FAA concentrations were similar between fast and slow growing children, Figure 5A,B; however, the observed differences were not statistically significant. In addition, there was a trend of higher free BCAA concentrations in the milk intended for fast growing children determined either by weight or head circumference; these trends were not statistically significant, Figure 4A and Figure 5A.

## 4. Discussion

The present results showed that HM intended for male infants had significantly higher concentrations of Glu, Gly, Cys, and Tyr and a trend for higher levels for free Gln and Ala compared with HM intended for females. Also, we found that there was a significantly greater abundance of free Glu and Ala in HM intended for faster growing infants. These results expand previous data and confirm that there are differences in FAA levels in HM intended for male infants and for those with greater weight gain regardless of gender.

High concentrations of FAA in HM is a distinctive trait present in humans. Free AA in HM are readily available and have potential effects in cells and tissues through specific receptors [19,20,21]. A recent report that studied the association between FAA in HM, total AA (free and bound amino acids, TAAs), and total protein levels with infant gender showed that breast-milk intended for infant females had higher protein and TAAs content during the first 3 months of lactation [14]. They also found higher levels of free Gln, Asn, Asp, Gly, Ser, and Glu, in milk for male infants, though these results were not statistically significant [14]. The authors concluded that AA of HM have differential composition and effects that are dependent on the infant’s sex and that their results need confirmation. Our linear multi-level regression model analysis that included exclusively FAA data from colostrum, transition, and mature milk (1 week through 4 months) showed statistically significant greater levels of Glu, Gly, Cys, and Tyr in HM of mothers with male infants; we also observed a trend for higher levels of Gln, Ala, Asn, Asp, and Ser in the milk for boys. The pattern of non-essential FAA observed in the present study was comparable to the one reported by van Sadelhoff et al. [14]. Similar to that study, we did not find statistically significant differences in the levels of essential FAA in HM for males or females. It is possible that the greater number of HM samples in the present report allowed a higher power for statistical analysis that permitted a clear distinction between FAA content in the milk intended for male and female infants.

Similar to other cohorts around the world, we previously reported that the levels of FAAs in colostrum, transition, and mature milk in Ecuadorian women increased with the time of lactation, with Tau, Glu, Gln, and Ala being most abundant in all stages of lactation [2,15,22]. Glutamate and glutamine contribute to gut development as energy substrates for intestinal epithelial cells, tight junction formation, and glutathione synthesis [10,11]; all these functions contribute to the establishment of an intestinal barrier for the growing infant. Also, glutamate and glutamine favor lymphocyte proliferation and differentiation including for those present in gut-associated lymphoid tissue [12,13]. In addition, glutamate together with glycine and cysteine, (more abundantly present in HM for boys) are part of glutathione that has a protective role against cell oxidative stress and improves immune function [23,24]. The rate-limiting factor for the synthesis of glutathione is the bioavailability of cysteine [23]. Considering that the body’s largest lymphoid component is located in the gut, it is possible that ample bioavailability of free Glu, Gln, Gly, and Cys contributes to infant growth by promoting gut-associated lymphoid tissue and intestinal development. The observed differences in FAA composition of HM reported here and by others could explain the differential effects on growth patterns of male and female infants [14]. In line with this idea, we also found greater concentrations of Glu and Ala in HM of children with faster weight gain compared with those that grew at slower pace; Gln concentrations were also higher although the differences were not significant. On the other hand, when we considered growth speed by head circumference measurements, the differences in Glu and Ala favored fast growing children although these were not statistically significant. Present results are in agreement with the positive association of free Gln with infant growth reported by Larnkjaer A, et al. [25].

Metabolome studies on breast milk that included measurements of FAA profiles from preterm infants indicate that BCAA and ITAA are more abundant in milk intended for faster growing children [9]. Here, we also found a non-statistically significant greater concentration of BCAA in HM of children with faster weight and head circumference gains compared with those that grew at slower pace. It is noticeable that the pattern of BCAA concentration in the present study closely resembled the observations reported previously in the faster growth group, with Val, Ile, and Leu being the most abundant in order of frequency [9]. It is possible that BCAA have positive effects on muscle protein anabolism [26] and contribute to fast growth of infants; BCAA stimulate the mTORC1 intracellular pathway associated with skeletal muscle protein synthesis [27]. Also, Leu, Gln, and Ala can favor anabolism through insulin release and growth functions improving infant’s growth [19]. These observations suggest that higher concentration of free BCAA are present in HM of infants with faster growth. It will be important to consider the present results in the context of clinical management of lactating infants with malnutrition and other pathologies. Further studies are needed to determine the underlying mechanisms that modify HM composition according to the gender and rate of growth of children. These results support the contention that FAA composition of HM could have differential effects on male and female infants. Milk composition, including its microbiome, varies depending on infant sex, time of parturition (term, preterm), and mother nutritional status. It is possible that endocrine, nervous, and immune control systems could module milk production of the mammary gland affecting the function of immune and milk producing epithelial cells [28].

We acknowledge that missing data points along the four months of the study could have limited the statistical power to find other possible significant differences. On the other hand, it is important to consider that data were missing at random. Hence, it is not likely that a potential bias was introduced in the present work.

## 5. Conclusions

This study showed that FAA composition of HM is associated with infants’ gender and their growth speed; our results demonstrated that free Glu, Gly, Cys, and Tyr in HM concentrations are higher in milk intended for boys. Also, there were greater concentrations of free Glu and Ala in HM in milk intended for faster growing children than for those infants with slower growth. Greater levels of these FAA in HM could favor gastrointestinal and immune development as well as promote anabolic metabolism in infants.

## Figures and Tables

**Figure 1 nutrients-11-02239-f001:**
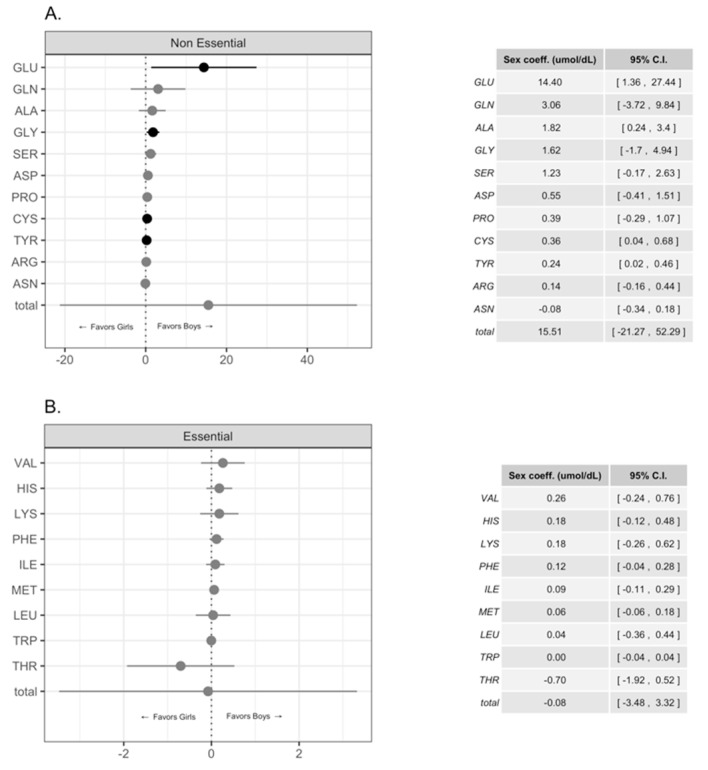
Differences in free non-essential (**A**) and essential (**B**) amino acid concentrations in human milk intended for female and male infants. Amino acid concentrations are expressed in mol/dL and coefficient estimates with their 95% confidence intervals are shown. Darker dots and lines indicate statistically significant differences.

**Figure 2 nutrients-11-02239-f002:**
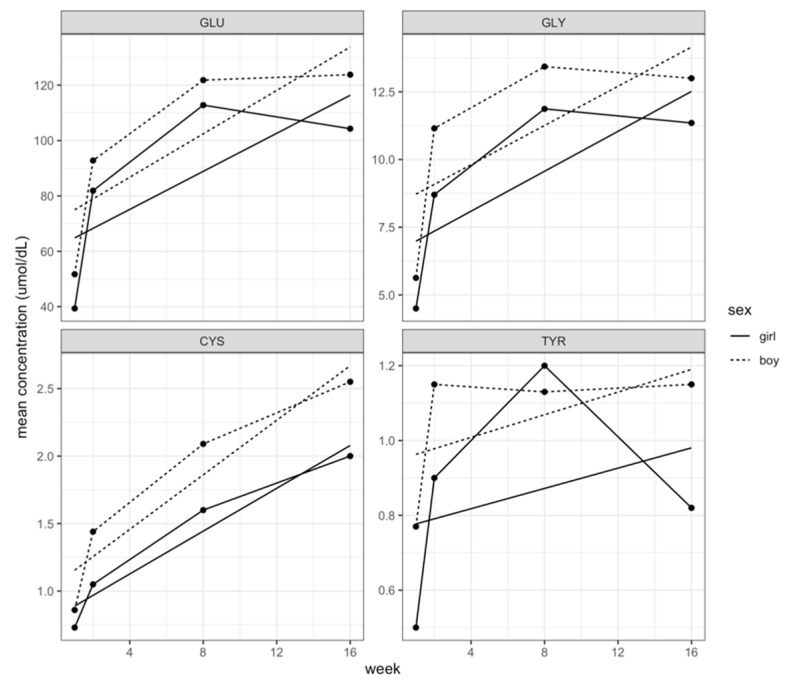
Measurements of free Glu, Gly, Cys, and Tyr concentrations (μmol/dL) in human milk for female and male suckling infants during the first 4 months of lactation. Straight lines represent a linear regression analysis of indicated free amino acids (FAA) concentrations over time by gender.

**Figure 3 nutrients-11-02239-f003:**
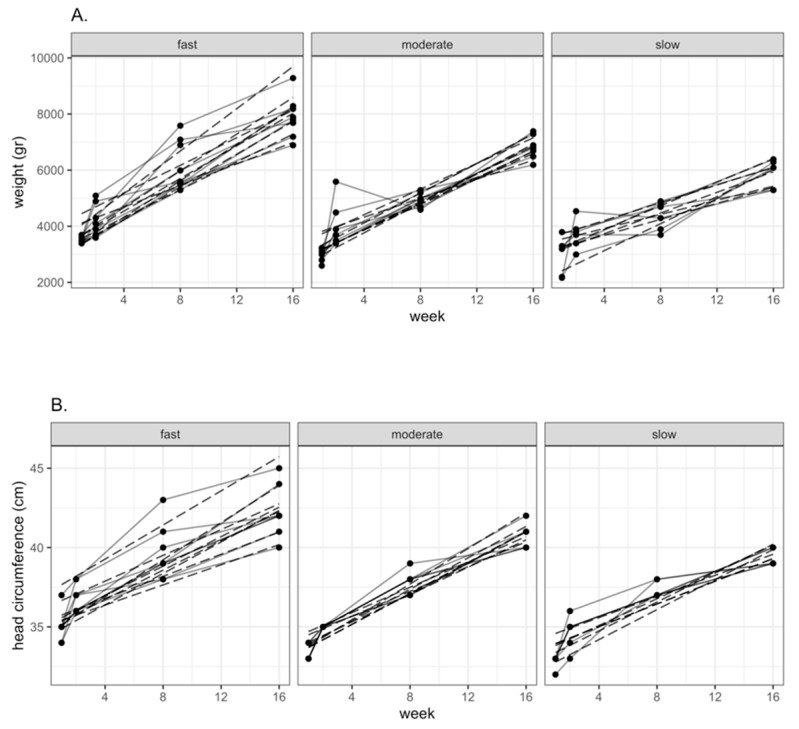
Patterns of weight (**A**) and head circumference (**B**) growth divided in tertiles of the studied infants. Dashed lines represent a linear estimation of individual growth and dots joined with continues lines are actual measurements used to estimate children’s growth. The slopes of these linear estimations were used to classify the population into tertiles of growth.

**Figure 4 nutrients-11-02239-f004:**
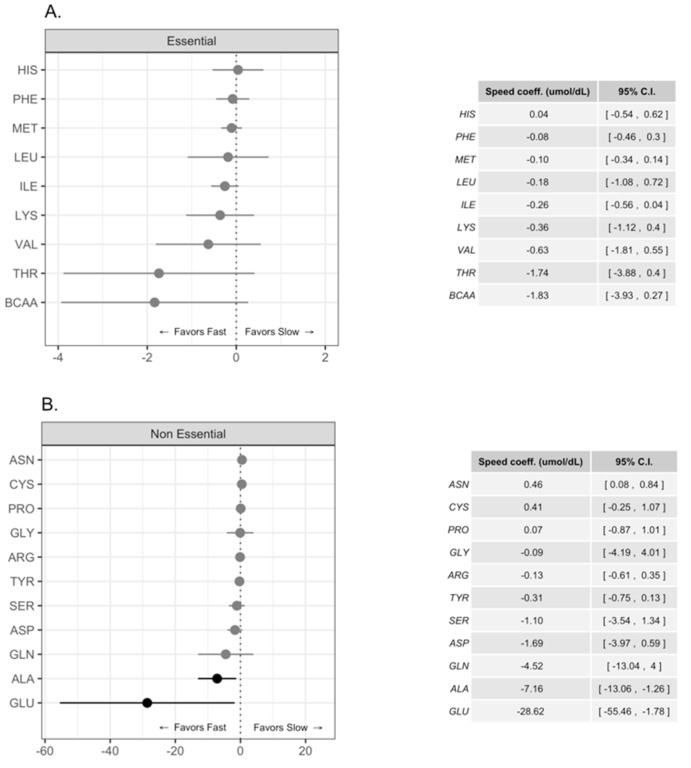
Differences in free essential (**A**) and non-essential (**B**) amino acid concentrations in human milk intended for fast and slow growing children by weight gain. Amino acid concentrations are expressed in mol/dL and coefficient estimates with their 95% confidence intervals are shown. BCAA = branched chain amino acids (Leucine, Isoleucine, and Valine). Darker dots and lines indicate statistically significant differences.

**Figure 5 nutrients-11-02239-f005:**
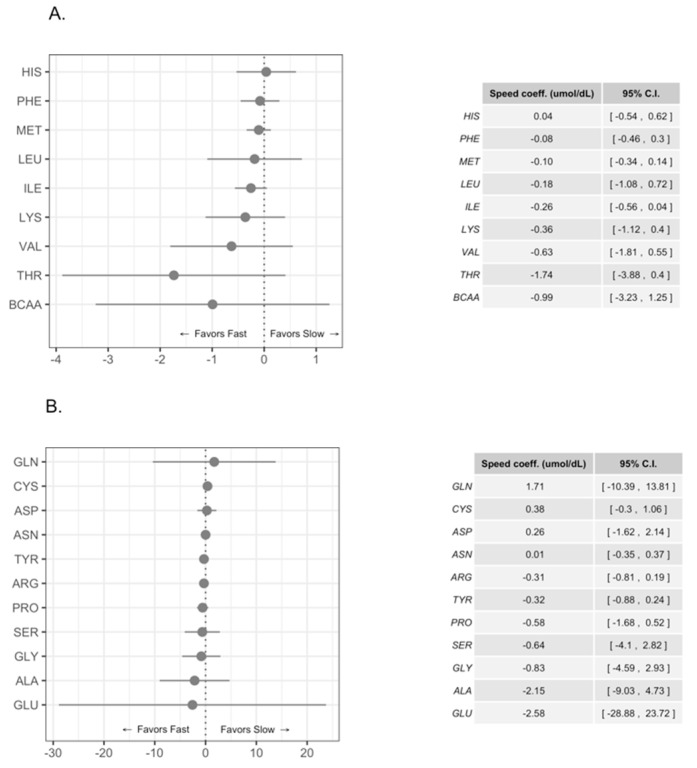
Differences in free essential (**A**) and non-essential (**B**) amino acid concentrations in human milk intended for fast and slow growing children by head circumference. Amino acid concentrations are expressed in mol/dL and coefficient estimates with their 95% confidence intervals are shown. BCAA = branched chain amino acids (Leucine, Isoleucine, and Valine).

**Table 1 nutrients-11-02239-t001:** Number of observations analyzed by time points of milk sample collection for all free amino acids.

**Week (Type of Milk)**	Number of Observations Used
**1 (colostrum, S1)**	61
**2 (transition milk, S2)**	47
**8 (mature milk, S3)**	38
**16 (mature milk, S4)**	37

**Table 2 nutrients-11-02239-t002:** Demographic and anthropometric characteristics of participating mothers and infants.

**Anthropometric Characteristics**		**Weight (kg)**	***p*** **Value**	**Height (m)**	***p*** **Value**
**Adolescent mothers (16.5 ± 1.0 years)**	**(*n* = 32)**	59.5 ± 8.1	0.815	1.6 ± 0.1	0.533
**Adult mothers (21.7 ± 2.6 years)**	**(*n* = 18)**	60.1 ± 9.7		1.6 ± 0.1	
**Infants Anthropometric Characteristics**		**Weight (kg)**	**Mean Difference**	**Head Circumference (cm)**	**Mean Difference**
**Female**	**Week 1**	3.45 ± 0.78 (*n* = 18)	3.41	33.53 ± 0.87 (*n* = 17)	6.02
	**Week 16**	6.86 ± 1.50 (*n* = 12)		39.55 ± 1.37 (*n* = 11)	
**Male**	**Week 1**	3.16 ± 0.43 (*n* = 32)	4.08	33.87 ± 1.36 (*n* = 32)	7.24
	**Week 16**	7.24 ± 1.09 (*n* = 17)		41.11 ± 1.57 (*n* = 18)	

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
