# Peer review of "Free Amino Acid Content in Human Milk Is Associated with Infant Gender and Weight Gain during the First Four Months of Lactation"

_nutrients, 2019, doi:10.3390/nu11092239_

Round 1

Reviewer 1 Report

Baldeón et al. tried to examine whether free amino acid (FAA) content in human milk was associated with infant sex and weight-gain pattern during the first four months of lactation in a small prospective cohort of primiparous Ecuadorian women. The authors reported that concentrations of Glu, Gly, Cys, and Tyr were higher in human milk for boys compared with that for girls. Additionally, they concluded that concentrations of free branched-chain amino acids in colostrum/transition milk were higher for faster weight gain neonates compared with their slower gain counterparts.

Major comments:

The authors did not show the number of final analyzed subjects (complete cases) for the results; thus, it was hard to judge whether the results were scientifically rigorous.  The results of 4 observational time points were drawn with several missing data; therefore, the conclusion may be biased.

The current study also included concerns for statistical analysis. According to the patterns of the changes of FAA concentrations over time (Fig 2), a linear regression (but potentially a quadratic regression) may not be appropriate.  The rationale for using a multilevel regression was unclear since the authors questioned the variation of FAA concentrations in the individual level (level 1).  In the result section, the results demonstrating statistical comparisons were not reported (e.g., beta coefficient and 95% CI or P-value).

Specific comments:

Abstract:

Please add beta and 95% CI of the main result.

Introduction:

Line 32:  “current recommendations for optimal child growth and development is exclusive breastfeeding”   No RCTs have proven that “exclusive” breastfeeding is superior to mixed/formula feeding; thus, “exclusive” should be dropped.

Methods:

Line 57-64: Please add the exclusion criteria and the number of excluded samples, and the number of the finally analyzed samples.

Line 65:  The sections of “2.2. Breast milk collection and amino acids measurements”. Observational time points should be defined and clarified. Also, means and SDs of each time points should be provided as the previous work of Ref. 15.

Line 84: Were babies/children naked? Did they wear light clothes?

Line 88: “At the time of recruitment, all but 10 of the children and all but 3 of the women were measured.”  In the final analyzed subjects, those with missing data should be excluded.

Line 90: The sections of “2.4. Ranking of infants according to growth rate”. Why were two different measurements of growth pattern applied? Should use one measurement.

Line 92: “the measurements over 1, 2, 8, and 16 weeks …”  Does this observational time point match to the time point of breast milk sampling?

Line 97: “As before”   Need reference number.

Line 101-104: “This prospective cohort study … on their children’s gender.”  These sentences should be moved to Section 2.1.

Line 108: “linear multi-level regression models with individual effects”  Please explain/clarify the rationale for multi-level regression analysis.

Result section:

The whole of the current results should be replaced to the result of complete case analysis. The current results are based on different analytic populations with variations of missing data, which probably lead to a biased inference.

Please show beta coefficients and 95% CIs and/or P-values when addressing statistically significance in the text. Most of the results for statistical significance were not available in the manuscript (e.g., Figure 2 without showing 95% CIs). Thus, any statistical conclusions may not be obtained with this current version.

Some unnecessary explanation parts (e.g., Line 129-131) should be transferred into Method section. Please elaborate to show results (but not methods) in the Result section.

Figures 1 and 4 should be shown in tables.

Tables 1 and 2 should be combined. Demonstration of the comparison between boys and girls is necessary.

Table 3: ANOVA across three groups (fast, intermediate, and slow) does not mean that the fast group is higher than the slow group. Also, the concentrations of FAA in colostrum and transition periods differed; thus, the combination of these two periods was biologically incorrect.

Figure and Table captions need elaborations. Hard to follow.

Line 195-196: This part is incorrect. Because of the prospective nature of the study, measurements from participants “are available” for all time points considered if the study is well-designed.  The variation of data points and the relatively large data missing imply that the quality of the study design was low. This would be a major limitation of the study.

Discussion:

Please add a discussion about the potential reason why the concentrations of FAA differ between boys and girls.

Line 218-220: “It is possible that the greater number of HM samples …”  This small study may not have enough statistical power for a robust conclusion. Thus, this discussion is confusing.

Conclusion:

Line 252-253: The first sentence of the conclusion does not correspond to the main research question, “Is free amino acid (FAA) content in human milk associated with infant sex?”

Author Response

Reviewer # 1

COMMENT: “Baldeón et al. tried to examine whether free amino acid (FAA) content in human milk was associated with infant sex and weight-gain pattern during the first four months of lactation in a small prospective cohort of primiparous Ecuadorian women. The authors reported that concentrations of Glu, Gly, Cys, and Tyr were higher in human milk for boys compared with that for girls. Additionally, they concluded that concentrations of free branched-chain amino acids in colostrum/transition milk were higher for faster weight gain neonates compared with their slower gain counterparts.

Major comments:

COMMENT 1: The authors did not show the number of final analyzed subjects (complete cases) for the results; thus, it was hard to judge whether the results were scientifically rigorous.  The results of 4 observational time points were drawn with several missing data; therefore, the conclusion may be biased.

RESPONSE: We thank the reviewer for the important comment. The number of observation for time point are summarized in the table below:

Week (type of milk)

Number of observations used

1 (colostrum, S1)

61

2 (transition milk, S2)

47

8 (mature milk, S3)

38

16 (mature milk, S4)

37

Consequently, there were at least 37 observations to fit the regression models. The details of the computations and the corresponding data and code can be found in the following link:

https://github.com/notblank/AA-leche-humana/blob/master/Missing%20at%20random.ipynb

During the study some patients missed collection dates (colostrum, sample 1-S1; transition S2, and mature breast milk S3 and S4). For those patients and time points of collection, data are missing. Nevertheless, we consider it is unlikely that the missing data could had led to a biased conclusion. To address this problem, we compared basal anthropometric measurements of mothers and children with complete and incomplete data and there were not statistically differences between the groups. In addition, since we were interested in the differences in the concentrations of FAA by sex, we analyzed the sex distribution for patients with missing and with complete data for each time point. After performing a Fisher exact test and a Score-type test we found that there is not evidence to state that these distributions are statistically different. Together this analysis demonstrates that the two groups are comparable and no bias was introduced considering data missing at random.

The following Table summarizes the indicated comparison:

Week (type of milk)

Data analyzed

Boys

Girls

Proportion of boys

P-value (Fisher)

P-value (Score)

2 (transition milk, S2)

Complete observations

27

20

0.57

Same proportion

Same proportion

Missing data

8

6

0.57

8 (mature milk, S3)

Complete observations

23

15

0.61

0.60

0.50

Missing data

12

11

0.52

16 (mature milk, S4)

Complete observations

20

17

0.54

0.60

0.54

Missing data

15

9

0.62

The table indicates the number of boys and girls per time point of study. The last two columns present the p-values of the two test that compare two binomial distributions.

The details of the computations and the corresponding data and code can be found in the following link:

https://github.com/notblank/AA-leche-humana/blob/master/Missing%20at%20random.ipynb”.

The original text has been modified to clarify this important point and the edited version in the 3. Results section, 3.1. Characteristics of study population now reads “During the study some patients missed collection dates (colostrum, sample 1-S1; transition S2, and mature breast milk S3 and S4). For those patients and time points of collection, data are missing. There were 50 mother infant pairs with complete anthropometric measurements at the beginning of the study. There were 15 mother infant pairs with missing data because anthropometry measurements were not performed or the sex of the infant was not recorded at base line.

In relation to milk samples, Table 1 summarizes the number of milk samples analyzed for time point of collection. There were at least 37 observations used to fit the regression models (see below). To address this problem, we compared basal anthropometric measurements of mothers and children with complete and incomplete data and there were not statistically differences between the groups. In addition, since we were interested in the difference in the concentration of FAA by sex, we analyzed the sex distribution for patients with missing and with complete data for each time point. After performing a Fisher exact test and a Score type-test we found that there were not differences in the analyzed groups with or without missing data. Together this analysis demonstrated that the two groups are comparable and no bias was introduced considering data missing at random.” Lines 136 – 150 in the revised text.

COMMENT 2: The current study also included concerns for statistical analysis. According to the patterns of the changes of FAA concentrations over time (Fig 2), a linear regression (but potentially a quadratic regression) may not be appropriate. The rationale for using a multilevel regression was unclear since the authors questioned the variation of FAA concentrations in the individual level (level 1).  In the result section, the results demonstrating statistical comparisons were not reported (e.g., beta coefficient and 95% CI or P-value).

RESPONSE: Although the patterns for the concentrations of the AA over time is not exactly linear, we used simple linear model, to summarize that the concentration pattern for one gender is above or below the other as in Figure 2.

For the most basic model we used:

???= α0 + α1????+ α2 ???+ αid

where FAA: is the free amino acid concentration; week: is the week at which samples were taken, sex: is the baby’s gender, and the last term represents individual effects. 

The sex coefficient of this simple regression model accounts for the difference between the concentrations for boys and girls after controlling for time and individual effects.

We used a multilevel regression model for two main reasons: 1. We wanted to compare our results with those reported by van Sadelhoff, J.H.J et al, 2018 (Reference 14 of the original manuscript); and 2. because these multilevel models give conservative estimations of the differences in the concentration of the FAA by sex since these models take into account possible unobserved individual effects. Indeed, in a simplified model without the individual effects:

???= α0 + α1????+ α2 ???

we found the same patterns and the same significant differences in FAA concentrations between males and females. The detailed computations for this simplified model can be found in the following link:

https://github.com/notblank/AA-leche-humana/blob/master/alpha_id%20effect.ipynb

The original text has been modified, we have included section 2.5.1 to explain the rational that support the used statistical models, the new text now reads: “2.5.1 Regression models

We used simple linear models to summarize differences in the concentrations of FAA over time between HM for boys and girls to summarize that the concentration pattern for one gender is above or below the other.

For the most basic model we used:

???= α0 + α1????+ α2 ???+ αid

where FAA: is the free amino acid concentration; week: is the week at which samples were taken, sex: is the baby’s gender, and the last term represents individual effects. 

The sex coefficient of this simple regression model accounts for the difference between the concentrations for boys and girls after controlling for time and individual effects.

We used a multilevel regression model because these multilevel models give conservative estimations of the differences in the concentration of the FAA by sex since these models take into account possible unobserved individual effects.

The multi-level regressions and plots were produced in R V3.6.0 using the lem4 and tidyverse packages. All the code and data for the regression analysis are available at https://github.com/notblank/AA-leche-humana”, lines 108 - 126

Specific comments:

Abstract:

COMMENT 3: Please add beta and 95% CI of the main result.

RESPONSE:  The beta and the 95% CI has been added in the abstract. The new text now reads: “We observed significantly higher concentrations of Glu 14.40 [1.35, 27.44], Gly 1.82 [0.24, 3.4], Cys 0.36 [0.03, 0.68], and Tyr 0.24 [0.02, 0.46] in HM intended for boys. Free Glu, Gly, Cys, and Tyr concentrations increased with time of lactation. In addition, there were higher concentrations of GLU 28.62 [1.78, 55.46] and ALA 7.16 [1.26, 13.06] in HM for children that presented faster weight gain than for those with slower gain. Conclusions: The present results showed that there are differences in FAA levels in HM intended for male and fast growing children.” Lines 17-23. In addition the new Figures 1,4, and 5 have also included a table with coefficients with their 95% CI.

Introduction:

COMMENT 3: Line 32:  “current recommendations for optimal child growth and development is exclusive breastfeeding”   No RCTs have proven that “exclusive” breastfeeding is superior to mixed/formula feeding; thus, “exclusive” should be dropped.

RESPONSE: Current recommendations for exclusive breastfeeding during the first 6 months of life is supported by several international agencies including the Wolrd Health Organization (https://www.who.int/elena/titles/exclusive_breastfeeding/en/). Thus, we would like to keep the original text. For ethical reasons it is not possible to compare in a RCT exclusive breast feeding versus non-exclusive breast feeding.

Methods:

COMMENT 4: Line 57-64: Please add the exclusion criteria and the number of excluded samples, and the number of the finally analyzed samples.

RESPONSE: We thank the reviewer for this important point. Lines 58-66 indicate the inclusion and exclusion criteria of the study, Section 2.1 Subjects, material and methods.

We have included the number of analyzed samples in the section 3.1. Characteristics of the study populations. We refer the reviewer to the RESPONSE to COMMENT 1.

COMMENT 5: Line 65: The sections of “2.2. Breast milk collection and amino acids measurements”. Observational time points should be defined and clarified. Also, means and SDs of each time points should be provided as the previous work of Ref. 15.

RESPONSE: We thank the reviewer for the comment, the original text has been modified to clarify the observational time points and time of sample collection. We also ask the reviewer to the RESPONSE to COMMENT 1. Amino acid concentrations were measured in mmol/dL and were previously published as indicated by the comment by the reviewer. The reviewer can read the data in the indicated publication, reference 15.

COMMENT 6: Line 84: Were babies/children naked? Did they wear light clothes?

RESPONSE: To clarify this point, we have added the following text “To record children’s weight, first the mother was weighed alone, the weight was recorded and subsequently the mother was weighed again holding her child wearing no cloths; the difference in weight between both measurements was registered as the weight of the child.” Lines 875-89 in the revised manuscript

COMMENT 7: Line 88: “At the time of recruitment, all but 10 of the children and all but 3 of the women were measured.” In the final analyzed subjects, those with missing data should be excluded.

RESPONSE: We agree with the reviewer, the missing data from the mothers and their children were no considered for the analysis. Please refer to section 3.1. Characteristics of the study population in the Results section.

COMMENT 8: Line 90: The sections of “2.4. Ranking of infants according to growth rate”. Why were two different measurements of growth pattern applied? Should use one measurement.

RESPONSE: We have considered the comment of the reviewer to use only one method to determine the pattern of growth of the study population for the analysis; we kept the division of the population in tertiles using the estimated slopes. The revised manuscript in section 2.4 now reads: “Infant population was divided in fast, intermediate, and slow growers using weight and head circumference at 1, 2, 8, and 16 weeks to estimate a slope and an intercept for each of the children. Then, the slopes and the intercepts were divided in tertiles. Children with a slope in the highest tertile were considered fast growers. This approach allowed assessment of the individual concentrations of FAA and the speed of child growth.

At the same time that the anthropometric measurements were taken, human milk samples were also collected, S1 to S4. To have a precise estimate of the speed of growth, we only used data from subjects whose FAA concentrations were measured at the four time points of the study”. Lines 94-102.

Using this method to divide the children according to the speed of growth, we estimated the differences on FAA concentrations in HM between fast and slow growing children according to weight and head circumference gains for all amino acids and for the sum of the three branched chain amino acids that have been associated with infant growth. We observed a statistically significant difference in GLU and ALA concentrations in favor of fast weight gaining children. These results are stated in 3.1.1 Differences in free amino acid concentrations in human milk for female and male infants subsection of Results. Lines 180-191. With this new analysis we complement the observed differences in FAA concentrations and children´s gender.

COMMENT 9: Line 92: “the measurements over 1, 2, 8, and 16 weeks …”  Does this observational time point match to the time point of breast milk sampling?

RESPONSE: Yes, the indicated time points are the same as the sampling points. The RESPONSE to COMMENT 8 clarifies this point.

COMMENT 10: Line 97: “As before”   Need reference number.

RESPONSE: The edition of the text in section 2.4. removed line 97 of the original text.

COMMENT 11: Line 101-104: “This prospective cohort study … on their children’s gender.”  These sentences should be moved to Section 2.1.

RESPONSE: We thank the reviewer for the comment and agree. We have moved the text as suggested by the reviewer. The revised text now reads “This prospective cohort study included 65 lactating mothers between 14 and 27 years of age that provided colostrum, transition, and mature milk during the first 4-months of lactation to determine FAA content; mothers and their children were recruited between 2009 and 2011. ”, Lines 58 – 60.

COMMENT 12: Line 108: “linear multi-level regression models with individual effects”  Please explain/clarify the rationale for multi-level regression analysis.

RESPONSE: We refer the reviewer to the Response of COMMENT 2 and section 2.5.1 in the modified text.

Result section:

COMMENT 13: The whole of the current results should be replaced to the result of complete case analysis. The current results are based on different analytic populations with variations of missing data, which probably lead to a biased inference.

RESPONSE: We refer the reviewer to the response to COMMENT 1 regarding the rationale behind how we treated the missing data.

COMMENT 14. Please show beta coefficients and 95% CIs and/or P-values when addressing statistically significance in the text. Most of the results for statistical significance were not available in the manuscript (e.g., Figure 2 without showing 95% CIs). Thus, any statistical conclusions may not be obtained with this current version.

RESPONSE: We have added the estimations with their 95% CI in the text and Figures.

COMMENT 15. Some unnecessary explanation parts (e.g., Line 129-131) should be transferred into Method section. Please elaborate to show results (but not methods) in the Result section.

RESPONSE: We thank the reviewer for the comment, we have edited the indicated lines to show only Results in this section. The edited version of the manuscript now reads: “We assessed the concentration of FAA in HM for male and female infants in our study group; we observed statistically significant higher concentrations of Glu, Gly, Cys, and Tyr and a trend for higher levels for free Gln and Ala in HM intended for boys, Figure 1 A, B.” Lines 165-167.

COMMENT 16. Figures 1 and 4 should be shown in tables.

RESPONSE: We have included the corresponding tables with the estimates and confidence intervals next to the coefficient plots. Figures 1, 4, and 5.

COMMENT 17. Tables 1 and 2 should be combined. Demonstration of the comparison between boys and girls is necessary.

RESPONSE: Attending the suggestion of the reviewer we have combined Tables 1 and 2 into one. We have also added the results of the comparison between male and female infants in the main text and the figure legend of the Table 2. The new text now reads “Table 2 shows the demographic and anthropometric characteristics of participating populations. Adolescent and adult mothers participated in this trial; none of them presented malnutrition at the time of recruitment. According to World Health Organization standards, on the first week after birth, 2% of children had abnormal low head circumference values; and 8% of children had low weight. While at week 16, 13.8% of children had abnormal low head circumference; and all infants had normal weights. As expected, male infants gained more weight and head circumference than female infants, Table 2. There were not statistically significant differences for weight and head circumferences between male and female infants at one week after birth (weight, p= 0.096; head circumference p= 0.349); at 16 weeks of age there were not differences in children´s weight (p= weight, p= 0.433); however, there were statistically differences in head circumference (p=0.011), Table 2.” Lines 154-163.

COMMENT 18. Table 3: ANOVA across three groups (fast, intermediate, and slow) does not mean that the fast group is higher than the slow group. Also, the concentrations of FAA in colostrum and transition periods differed; thus, the combination of these two periods was biologically incorrect.

RESPONSE: In the reviewed version of the manuscript, Table 3 was eliminated. Please refer to the RESPONSE to COMMENT 8.

COMMENT 19. Figure and Table captions need elaborations. Hard to follow.

RESPONSE: We have modified the legends of Figures and Tables to clarify this point.

COMMENT 20. Line 195-196: This part is incorrect. Because of the prospective nature of the study, measurements from participants “are available” for all time points considered if the study is well-designed. The variation of data points and the relatively large data missing imply that the quality of the study design was low. This would be a major limitation of the study.

RESPONSE: We thank the reviewer for the comment. We have removed section 3.3 Formatting of Mathematical Components and we have added section 2.5.1 which no longer contains the text of this COMMENT, and edited section 3.1. where we explain how we handle the missing data. Also, please refer to COMMENT 1.

Discussion:

COMMENT 21. Please add a discussion about the potential reason why the concentrations of FAA differ between boys and girls.

RESPONSE: The objective of the present study was to assess differences on FAA composition of HM intended for male and female infants. However, we have included new text to clarify this comment. “Milk composition, including its microbiome, varies depending on infant sex, time of parturition (term, preterm), and mother nutritional status. It is possible that endocrine, nervous, and immune control systems could module milk production of the mammary gland affecting the function of immune and milk producing epithelial cells (27).” Lines 284-287.

COMMENT 22. Line 218-220: “It is possible that the greater number of HM samples …”  This small study may not have enough statistical power for a robust conclusion. Thus, this discussion is confusing.

RESPONSE: The sample size of the present study is greater than two studies published in Nutrient regarding human milk composition and infants gender. Please refer to references (9 and 14) of the new version of the manuscript. As indicated in in Response to COMMENT 1, there were at least 37 observations per time point of milk sample analysis. Based on these number of samples we stated that “the greater number of HM samples in the present report allowed a higher power for statistical analysis that permitted a clear distinction between FAA content in the milk intended for male and female infants.” We consider that this statement is valid. Also, we have added a paragraph where we acknowledge the limitations of the study where we also clarify this point, Lines 245-247. In addition, we have added a paragraph that acknowledges the limitations of our study regarding the number of HM samples used. Lines 288-291.

Conclusion:

COMMENT 23. Line 252-253: The first sentence of the conclusion does not correspond to the main research question, “Is free amino acid (FAA) content in human milk associated with infant sex?”

RESPONSE: We agree with the comment and the original text has been edited.  The Conclusions section now reads: “This study showed that FAA composition of HM is associated with infants’ gender and their growth speed; our results demonstrated that free Glu, Gly, Cys, and Tyr in HM concentrations are higher in milk intended for boys. Also, there was greater concentrations of free Glu and Ala in HM in milk intended for faster growing children than for those infants with slower growth. Greater levels of these FAA in HM could favor gastrointestinal and immune development as well as promote anabolic metabolism in infants.” Lines 293-298.

Reviewer 2 Report

Introduction too laconic, not very detailed, lack of information regarding the concentration of free amino acids in human milk, their significance as well as the functional role in the early period of human life.

In the part,  materials and methods  the study group have not been described Tables 1 and 2 should be moved from the part of the results to the materials and methods.

Table 2 is difficult to analyze, it must be reedited,  the descriptions of the gender of children must be changed to male and female. weight gain and Growth of newborns and infants should be evaluated by using percentile charts, for example. WHO. 

Table 3. Concentrations of free branched chain amino acids -values are given in kg and cm, it is necessary to re-edit the table, the current version is misleading and incomprehensible

lack - in the references and also in discussion - several important publications concerning free amino acids in breastmilk as well as optimal weight and growth gains, among them:

1.J Nutr Biochem. 2017 Mar;41:1-11. doi: 10.1016/j.jnutbio.2016.06.001.

2. J Pediatr Gastroenterol Nutr. 2016 Sep;63(3):374-8. doi: 10.1097/MPG.0000000000001195

3. PLoS One. 2018 Jun 1;13(6):e0197713. doi: 10.1371/journal.pone.0197713.

language necessarily requires correction, often colloquial rather than scientific terms are used.

Author Response

Reviewer 2.

Comments and Suggestions for Authors

COMMENT 1: Introduction too laconic, not very detailed, lack of information regarding the concentration of free amino acids in human milk, their significance as well as the functional role in the early period of human life.

RESPONSE: We used the text in the introduction to set up the stage and provide a general idea of the reason why we carried out the study. In the Discussion section we detail the significance as well as the functional role of FAA in the early development of infants. We refer the reviewer to Lines 248-278.

COMMENT 2: In the part, materials and methods  the study group have not been described Tables 1 and 2 should be moved from the part of the results to the materials and methods.

RESPONSE: The original text has been modified according to the reviewer´s COMMENT 2. The revised text now reads “2.1. Subjects, material and methods

            This prospective cohort study included 65 lactating mothers between 14 and 27 years of age that provided colostrum, transition, and mature milk during the first 4-months of lactation to determine FAA content; mothers and their children were recruited between 2009 and 2011.

Participating mothers were recruited after their child´s delivery at Hospital Gíneco-Obstétrico Isidro Ayora in Quito, which serves low-income families. We invited to participate 65 healthy primiparous women from Quito who gave birth to a single, at-term healthy infant without obstetric complications, and who indicated their willingness to feed their child exclusively with their breast-milk for at least 4 months. Women with history of abortions or infections during pregnancy were not included in the study [15].”, Lines 58-66.

Tables 1 and 2 were combined in a single Table 2; since this table describes the results of the recruited population, data are presented in the Result section. Lines 154-163.

COMMENT 3: Table 2 is difficult to analyze, it must be reedited, the descriptions of the gender of children must be changed to male and female. weight gain and Growth of newborns and infants should be evaluated by using percentile charts, for example. WHO.

RESPONSE: We thank the reviewer for the comments, regarding Table 2, following this reviewer’s comment and from another referee, Table 1 and 2 have been combined and edited for clarity. Line 222. The new text to present the results considering WHO standards nor reads: “According to World Health Organization standards, on the first week after birth, 2% of children had abnormal low head circumference values; and 8% of children had low weight. While at week 16, 13.8% of children had abnormal low head circumference; and all infants had normal weights. As expected, male babies gained more weight and head circumference than female infants, Table 2. There were not statistically significant differences for weight and head circumferences between male and female infants at one week after birth (weight, p= 0.096; head circumference p= 0.349); at 16 weeks of age there were not differences in children´s weight (p= weight, p= 0.433); however, there were statistically differences in head circumference (p=0.011), Table 2.” Lines 156-163.”

COMMENT 4: Table 3. Concentrations of free branched chain amino acids -values are given in kg and cm, it is necessary to re-edit the table, the current version is misleading and incomprehensible

RESPONSE: In the new version of the manuscript Table 3 was eliminated. The association of FAA and BCAA with the speed of infants’ growth is now expressed in Figures 4 and 5.

COMMENT 5: lack - in the references and also in discussion - several important publications concerning free amino acids in breastmilk as well as optimal weight and growth gains, among them:

1.J Nutr Biochem. 2017 Mar;41:1-11. doi: 10.1016/j.jnutbio.2016.06.001.

J Pediatr Gastroenterol Nutr. 2016 Sep;63(3):374-8. doi: 10.1097/MPG.0000000000001195

PLoS One. 2018 Jun 1;13(6):e0197713. doi: 10.1371/journal.pone.0197713.

RESPONSE: We did not use reference 1 because it analyses total amino acid composition of breast milk while our interest is the study of FAA. A priory their behavior during lactation is different, for instance, data indicate that total amino acids concentration decrease through lactation while FAA concentrations increase. These could reflect different biological activities of bound and free amino acids. Regarding reference 3, this report does not measure FAA concentrations in human milk rather the measurements were done on infants’ serum. Hence, the objectives of both studies are different.

We have included reference 2 in the Discussion section. The new text reads: “Present results are in agreement with the positive association of free Gln with infant growth reported by Larnkjaer A, et al [25]” Lines, 267-268.

COMMENT 6: language necessarily requires correction, often colloquial rather than scientific terms are used.

RESPONSE: The manuscript has been edited by native English-speakers.

Reviewer 3 Report

Free amino acid content in human milk is associated with infant gender and weight gain during the first four months of lactation

Authors

Manuel E. Baldeon * , Federico Zertuche , Nancy Flores , Marco Fornasini

Comments to the Authors

This was a prospective cohort study that assessed the concentration of breast milk FAA during the first 4 months of lactation according to the gender of infants and growth pattern.

The authors observed an higher concentrations of Glu, Gly, Cys, and Tyr in HM for boys. Free Glu, Gly, Cys, and Tyr concentrations increased with time of lactation. In addition, there were significantly higher concentrations of free branched-chain amino acids (BCAA) in colostrum and transition milk for children that presented faster weight gain than for those with slow gain.  

The study is of potential interest although needs major revisions.

Major comments

Material and methods

1. The author have to clarify the reason way they design to collect the milk sample after the night fasting period. I imagine that it is due to the influence of the diet in modulating the FAA concentration. If it is true, the author have to provide data regarding the diet of the nurses. Furthermore it must be considered that, in order to better evaluate the milk composition, it would be necessary to collect an extracted sample from a 24-hour pool, thatbetter reflects the actual infant’s intake. 2. The methods used to measure the infants weight is certainly inaccurate and could be represent a bias and they have to relate  the data to a growth curve (I suggest the WHO Charts). In addition it is quite unclear the methods used to categorize the tertiles. The author could use the growth rate (for example g/weeks or kg/months). 3. Can the authors provide data regarding the total proteins concentration during lactation according with sex? It is possible that male infants have not only a major amount of non essential AA but also an higher amount of total proteins and it can be responsible of and higher growth rate. 4. The authors enrolled both adolescent and adult mothers. There was any difference in the expression of FAA? The authors should include data according with the demographic characteristics of participating mothers.  

Minor comments:

1. Lines 129-131. This sentence is the aim of the study. The authors have to insert the aim at the end of introduction section. 2. Lines 143-145. These are comments,please insert in the discussion section. 3. Figure 2 is difficult to understand. Specifically I can’t understand the difference between the dashed lines and straight lines. 4. Figure 3A (moderate and slow). There are few subjects with a peculiar pattern of growth. Did someone have a weight loss since baseline to 8 weeks? 5. Tables 2. Why the rows “birth weight” and “head circumference” are centered? The numbers below do not only  show data regarding birth. 6. Table 3. The caption of the table explained that the data showed are AA levels in μmol/dL. Why the table shows data regarding weight and head circumpherence?   7. Lines 247-248. How the data obtained can have future clinical impact?

Author Response

Reviewer 3.

Comments to the Authors

This was a prospective cohort study that assessed the concentration of breast milk FAA during the first 4 months of lactation according to the gender of infants and growth pattern.

The authors observed an higher concentrations of Glu, Gly, Cys, and Tyr in HM for boys. Free Glu, Gly, Cys, and Tyr concentrations increased with time of lactation. In addition, there were significantly higher concentrations of free branched-chain amino acids (BCAA) in colostrum and transition milk for children that presented faster weight gain than for those with slow gain. 

COMMENT 1: The study is of potential interest although needs major revisions.

RESPONSE: We thank the reviewer for the comment that validates current work.

Major comments

Material and methods

COMMENT 2: 1. The author have to clarify the reason way they design to collect the milk sample after the night fasting period. I imagine that it is due to the influence of the diet in modulating the FAA concentration. If it is true, the author have to provide data regarding the diet of the nurses. Furthermore it must be considered that, in order to better evaluate the milk composition, it would be necessary to collect an extracted sample from a 24-hour pool, thatbetter reflects the actual infant’s intake.

RESPONSE: Milk samples were collected after night fasting condition because in the clinical practice, biological samples are collected under basal conditions which also accounts for circadian rhythms. Under basal conditions it is possible to avoid the potential differences in biological markers due to meal consumption.

COMMENT 3: 2. The methods used to measure the infants weight is certainly inaccurate and could be represent a bias and they have to relate  the data to a growth curve (I suggest the WHO Charts). In addition it is quite unclear the methods used to categorize the tertiles. The author could use the growth rate (for example g/weeks or kg/months).

RESPONSE: We thank the reviewer for the comment. We related our data to WHO growth standard curves. Lines, 156-163. Also, we have simplified the original text by only retaining the method in which we used estimated slopes (this method uses the units you indicate: g/wks or cm/wks) to assess the speed of growth to classify the population. Lines, 94-102; See also subsection 3.1.1 Differences in free amino acid concentrations in human milk for female and male babies of Results. Lines 180-189.

COMMENT 4: 3. Can the authors provide data regarding the total proteins concentration during lactation according with sex? It is possible that male infants have not only a major amount of non essential AA but also an higher amount of total proteins and it can be responsible of and higher growth rate.

RESPONSE: We thank the reviewer for the comment. In the present study we only measured the concentrations of free amino acids. We do not have data on total protein concentrations since this was not the objective of the study.

COMMENT 5: 4. The authors enrolled both adolescent and adult mothers. There was any difference in the expression of FAA? The authors should include data according with the demographic characteristics of participating mothers.

RESPONSE: In a previous report we showed that FAA concentrations in human breast milk change over the first four months of lactation and were not related to mother’s age (see reference 15 of present manuscript).

Minor comments:

COMMENT 6: 1. Lines 129-131. This sentence is the aim of the study. The authors have to insert the aim at the end of introduction section.

RESPONSE: We agree with the reviewer and the last sentence of the introduction now reads: “Here we complement our previous work and report the relationships between FAA content in HM of Ecuadorian women (colostrum, sample 1-S1; transition S2, and mature breast milk S3) and their children’s gender and pattern of growth. Lines 52-55.

COMMENT 7: 2. Lines 143-145. These are comments, please insert in the discussion section.

RESPONSE: We introduced the rationale of the results that we are presenting with the indicated sentence. We also discussed the indicated article in the Discussion.

COMMENT 8: 3. Figure 2 is difficult to understand. Specifically I can’t understand the difference between the dashed lines and straight lines.

RESPONSE: Figure 2 has been modified for clarity. The new legend now reads: “Figure 2. Measurements of free Glu, Gly, Cys, and Tyr concentrations (μmol/dL) in human milk for female and male suckling infants during the first 4 months of lactation. Straight lines represent a linear regression analysis of indicated FAA concentrations over time by gender.” Also the Figure contains a key that distinguishes boys dashed and girls continuous lines.

COMMENT 9: 4. Figure 3A (moderate and slow). There are few subjects with a peculiar pattern of growth. Did someone have a weight loss since baseline to 8 weeks?

RESPONSE: Yes, as shown in the figures, some individual lost weight between weeks 2 and 8. Nevertheless, all infants had normal weight at week-16. Lines 156-160.

COMMENT 10: 5. Tables 2. Why the rows “birth weight” and “head circumference” are centered? The numbers below do not only  show data regarding birth.

RESPONSE: To clarify this point we have merge and edited Tables 1 and 2 into a single Table 2. Line 222.

COMMENT 11: 6. Table 3. The caption of the table explained that the data showed are AA levels in μmol/dL. Why the table shows data regarding weight and head circumpherence? 

RESPONSE: In the new version of the manuscript Table 3 was eliminated. The association of FAA and BCAA with the speed of infants’ growth is now expressed in Figures 4 and 5.

COMMENT 12: 7. Lines 247-248. How the data obtained can have future clinical impact?

RESPONSE: Present results can have clinical applications in the treatment of undergrowth children. For example, it will be important to analyze whether HM rich in Glu, Ala, Gln help to accelerate preterm and undernourished infants´ recovery.

Round 2

Reviewer 1 Report

The authors made revisions appropriately — no further comment.

Reviewer 3 Report

The authors respond adequately and comprehensively to the questions posed by the reviewer, I therefore consider that the manuscript can be accepted in the present form.